# Clinical usefulness of serum autotaxin levels for predicting decompensation development and prognosis in patients with compensated cirrhosis

**Chisato Saeki**[1,2]*, **Tsunekazu Oikawa**[1], **Tomoya Kanai**[1,2], **Sachie Kiryu**[1,2], **Hiroshi Kamioka**[1], **Kaoru Ueda**[1], **Masanori Nakano**[1,2], **Yuichi Torisu**[1,2], **Masayuki Saruta**[1], **Akihito Tsubota**[3]*

**1** Division of Gastroenterology and Hepatology, Department of Internal Medicine, The Jikei University School of Medicine, Minato-ku, Tokyo, Japan, **2** Division of Gastroenterology, Department of Internal Medicine, Fuji City General Hospital, Fuji-shi, Shizuoka, Japan, **3** Project Research Units, Research Center for Medical Science, The Jikei University School of Medicine, Minato-ku, Tokyo, Japan

* chisato@jikei.ac.jp (CS); atsubo@jikei.ac.jp (AT)

## Abstract

### Aim

The progression from compensated to decompensated cirrhosis markedly worsens prognosis. This study investigated the clinical utility of serum autotaxin (ATX) levels as a predictive biomarker for decompensation and mortality, particularly in patients with compensated cirrhosis.

### Methods

A total of 210 patients with cirrhosis (165 with compensated cirrhosis) were retrospectively analyzed and classified into three groups based on baseline serum ATX levels: low-, intermediate-, and high-ATX groups.

### Results

Over a median follow-up of 51.6 months, 43 patients (20.5%) died of liver-related events, and 26 patients (15.8%) with compensated cirrhosis at baseline progressed to decompensation. In both the overall cohort and the compensated cirrhosis subgroup, the high-ATX group had significantly lower cumulative survival rates than the intermediate- and low-ATX groups ($p < 0.001$ for both). Multivariate analysis identified high serum ATX levels as an independent predictor of mortality (overall cohort: hazard ratio [HR], 1.661; $p = 0.039$; compensated cirrhosis subgroup: HR, 7.488; $p < 0.001$). Furthermore, the cumulative incidence of decompensation was highest in the high-ATX group ($p < 0.001$), and high serum ATX levels were independently associated with an increased risk of decompensation (HR, 5.502; $p < 0.001$).

**Data availability statement:** The dataset used in this study contains potentially identifiable and sensitive patient information and therefore cannot be made publicly available due to ethical restrictions imposed by the Ethics Committees of The Jikei University School of Medicine and Fuji City General Hospital. However, de-identified data may be made available upon reasonable request from qualified researchers who meet the criteria for access to confidential data. Requests for access to the data should be directed to the Ethics Committee of The Jikei University School of Medicine (email: rinri@jikei.ac.jp).

**Funding:** The author(s) received no specific funding for this work.

**Competing interests:** The authors have declared that no competing interests exist.

## Conclusion

ATX represents a promising non-invasive biomarker for predicting decompensation and poor prognosis in patients with compensated cirrhosis.

---

## Introduction

Cirrhosis is a leading cause of global morbidity and mortality, accounting for over one million deaths annually [1,2]. The estimated annual rate of progression from compensated cirrhosis (asymptomatic phase) to decompensated cirrhosis (characterized by the development of ascites, encephalopathy, variceal bleeding, and jaundice) is 5%–7% [3]. Patients with decompensated cirrhosis have a much worse prognosis than those with compensated cirrhosis (median survival: < 2 years and > 12 years, respectively) [3]. Therefore, identifying patients at risk for decompensation and implementing early therapeutic intervention are crucial for reducing mortality in this population.

The Child–Pugh (CP) classification, which incorporates the severity of encephalopathy and ascites, along with serum albumin, total bilirubin, and prothrombin time (PT) levels, is commonly used to assess hepatic functional reserve [4]. This classification system estimates the prognosis of cirrhosis, with mortality rates rising as the disease stage progresses from CP class A to class B or C [3]. The model for end-stage liver disease (MELD) score, calculated based on serum total bilirubin, creatinine, and international normalized ratio (INR) of PT levels, was originally developed to predict short-term mortality in patients with recurrent variceal bleeding or refractory ascites undergoing transjugular intrahepatic portosystemic shunt [5]. It is now widely used to prioritize candidates for liver transplantation. However, in patients with compensated cirrhosis, both CP and MELD scores tend to cluster within the lower range due to normal or mildly abnormal biochemical values, which limits their prognostic utility [6,7]. Furthermore, neither scoring system was specifically designed to predict the progression to decompensation [8]. Therefore, there is a critical need for a simple, routinely measurable biomarker that can reliably predict disease progression, phase transition, and prognosis in patients with compensated cirrhosis.

Autotaxin (ATX), a secreted glycoprotein encoded by the ectonucleotide pyrophosphatase/phosphodiesterase 2 gene, was initially identified in human melanoma cell cultures [9]. ATX catalyzes the conversion of lysophosphatidylcholine into lysophosphatidic acid (LPA), a bioactive phospholipid involved in various physiological processes, including the proliferation and contractile activity of hepatic stellate cells, which are key contributors to hepatic fibrogenesis [10,11]. Circulating ATX is primarily degraded by hepatic sinusoidal endothelial cells; therefore, hepatic fibrosis and the resulting endothelial dysfunction reduce ATX clearance, leading to increased serum ATX levels [11,12]. Serum ATX levels have been shown to correlate with hepatic fibrosis stage, disease severity, and adverse clinical outcomes in various chronic liver diseases, including hepatitis B and C, cholestatic liver diseases (such as primary biliary cholangitis [PBC] and primary sclerosing cholangitis [PSC], and metabolic

dysfunction-associated steatotic liver disease (MASLD) [12–19]. However, the utility of ATX in predicting the development of decompensation and long-term prognosis, specifically in patients with compensated cirrhosis, remains unclear.

This study aimed to investigate whether serum ATX levels can serve as a predictive biomarker for hepatic decompensation and long-term clinical outcomes, particularly in patients with compensated cirrhosis.

## Materials and methods

### Study participants

This retrospective study involved 210 consecutive patients with cirrhosis who were managed at two hospitals, the Jikei University School of Medicine (Tokyo, Japan) and Fuji City General Hospital (Shizuoka, Japan), between July 2019 and December 2023.

Data were accessed for research purposes at Jikei University School of Medicine between 16/05/2022 and 16/07/2025 and at Fuji City General Hospital between 15/08/2022 and 25/07/2025. All data were fully anonymized before the authors accessed them. The authors did not have access to information that could identify individual participants during or after data collection. The inclusion criteria were as follows: (i) cirrhosis of any etiology and (ii) availability of baseline serum ATX measurements. The exclusion criteria were as follows: (i) preexisting malignancies, including hepatocellular carcinoma (HCC); (ii) acute liver failure; (iii) a history of liver transplantation; and (iv) current dialysis treatment. Cirrhosis was diagnosed based on laboratory data, endoscopic findings (i.e., esophageal/gastric varices), and characteristic imaging features (e.g., surface nodularity, liver deformity with right lobe atrophy and lateral segment/caudate lobe hypertrophy, splenomegaly, and ascites) [20]. Decompensated cirrhosis was defined by the presence of ascites, encephalopathy, variceal hemorrhage, and/or jaundice [21]. Hepatic functional reserve was assessed using the CP, MELD, and albumin-bilirubin (ALBI) scores [4,5,22]. HCC was diagnosed according to the American Association for the Study of Liver Diseases guidelines [23]. Patients were followed regularly in the outpatient clinic at 1–2 month intervals, with laboratory tests performed at each visit. Imaging examinations, including computed tomography, magnetic resonance imaging, and/or ultrasonography, were conducted approximately every 4 months to assess ascites and screen for HCC. Endoscopic examinations for variceal surveillance were performed every 6–12 months. The primary endpoint was liver-related mortality; deaths from non-liver-related causes were treated as censored observations. This study adhered to the 2013 Declaration of Helsinki's ethical principles and was approved by the ethics committees of the Jikei University School of Medicine (approval number: 34−021) and Fuji City General Hospital (approval number: 279). Informed consent was obtained from participants via an opt-out procedure, and the requirement for written informed consent was waived by the ethics committees.

### Laboratory assessments

All laboratory parameters, including total bilirubin, albumin, creatinine, sodium, PT–INR, and platelet count, were measured using standardized protocols. Serum ATX levels were determined by a two-site enzyme immunoassay employing the automated immunoassay analyzer AIA-2000 (Tosoh, Tokyo, Japan).

### Patient classification based on baseline serum ATX levels

Patients were categorized into three groups based on serum ATX levels: low (<first quartile), intermediate (interquartile range), and high (> third quartile) (S1 Fig). Furthermore, given that serum ATX levels are reportedly higher in women than in men [24], sex-stratified analyses were conducted using sex-specific ATX cutoff values.

### Statistical analysis

Continuous variables are presented as medians with interquartile ranges, and categorical variables as frequencies with corresponding percentages. Between-group differences were assessed using the Mann–Whitney U test or Kruskal–Wallis

test for continuous variables, and the chi-square test for categorical variables. Correlations between serum ATX levels and baseline clinical parameters were evaluated using Spearman's rank correlation coefficient. Cumulative survival and decompensation incidence rates were estimated using the Kaplan–Meier method, and between-group differences were evaluated using the log-rank test. Univariate Cox proportional hazards models were employed to identify factors associated with mortality and the development of decompensation. Variables with $p < 0.1$ in the univariate analysis were entered into the final multivariate Cox proportional hazards models (using forward-stepwise method). Age and sex were included in multivariate analyses as clinically relevant covariates regardless of their significance in univariate analyses. Total bilirubin, albumin, PT-INR, and creatinine, which constitute components of the CP, MELD, and ALBI scores, were not entered simultaneously into the multivariate models to avoid multicollinearity. All statistical analyses were conducted using SPSS Statistics version 27.0 (IBM Japan, Tokyo, Japan). A $p$-value of < 0.05 was considered statistically significant.

## Results

### Patient characteristics

The baseline characteristics of the 210 participants (130 men [61.9%] and 80 women [38.1%]; median age, 68.0 [58.0–78.0] years) are summarized in Table 1. The prevalence of decompensated cirrhosis was 21.4% (45/210). The median CP, MELD, and ALBI scores were 5 (5–6), 8 (6–10), and −2.65 (−2.90 to −2.18), respectively.

### Comparison of clinical characteristics between men and women

Female patients had significantly higher ATX levels ($p < 0.001$) and lower MELD score ($p = 0.044$) and creatinine levels ($p < 0.001$) compared to male patients (S1 Table). Alcohol-related liver disease manifested more frequently in men, while other etiologies predominated in women ($p < 0.001$).

**Table 1. Clinical characteristics of the three groups based on serum autotaxin levels.**

| Variable | ALL | L-ATX | I-ATX | H-ATX | p value |
|---|---|---|---|---|---|
| Patients, n (%) | 210 | 52 (24.8) | 106 (50.5) | 52 (24.8) | |
| Men/Women, n (%) | 130 (61.9)/80 (38.1) | 42 (80.8)/10 (19.2) | 63 (59.4)/43 (40.6) | 25 (48.1)/27 (51.9) | 0.002 |
| Age (years) | 68.0 (58.0–78.0) | 67.0 (56.5–75.8) | 70.5 (61.0–79.0) | 67.5 (56.3–77.8) | 0.312 |
| Etiology | | | | | |
| HBV/HCV/alcohol/other, n | 21/48/78/70 | 7/9/18/18 | 9/20/37/40 | 5/12/23/12 | 0.599 |
| Decompensated cirrhosis, n (%) | 45 (21.4) | 0 (0.0) | 16 (15.1) | 29 (55.8) | < 0.001 |
| Child-Pugh score | 5 (5–6) | 5 (5–5) | 5 (5–6) | 7 (5–9) | < 0.001 |
| MELD score | 8 (6–10) | 7 (6–8) | 8 (6–10) | 11 (7–14) | < 0.001 |
| ALBI score | −2.65 (−2.90−−2.18) | −2.86 (−3.07−−2.65) | −2.65 (−2.91−−2.29) | −2.00 (−2.34−−1.36) | < 0.001 |
| Total bilirubin (mg/dL) | 0.8 (0.6–1.2) | 0.8 (0.5–0.9) | 0.8 (0.6–1.1) | 1.3 (0.9–2.2) | < 0.001 |
| Albumin (g/dL) | 3.9 (3.5–4.2) | 4.2 (4.0–4.4) | 4.0 (3.6–4.2) | 3.3 (2.9–3.8) | < 0.001 |
| Prothrombin time INR | 1.07 (1.00–1.19) | 1.00 (1.00–1.08) | 1.07 (1.00–1.16) | 1.20 (1.07–1.35) | < 0.001 |
| Creatinine (mg/dL) | 0.8 (0.7–1.1) | 0.9 (0.7–1.1) | 0.8 (0.7–1.1) | 0.8 (0.7–1.0) | 0.601 |
| Sodium (mEq/L) | 140 (138–142) | 141 (140–143) | 140 (139–142) | 139 (137–140) | < 0.001 |
| Platelet (x10⁴/μl) | 12.6 (9.2–16.4) | 14.6 (12.2–20.1) | 12.4 (9.3–15.3) | 9.8 (6.5–14.1) | < 0.001 |
| Autotaxin (mg/L) | 1.377 (1.028–1.763) | 0.879 (0.756–0.957) | 1.377 (1.175–1.525) | 2.197 (1.890–2.827) | < 0.001 |

Continuous variables are shown as median (interquartile range). Statistical analysis was performed using the chi-squared test or the Kruskal-Wallis test, as appropriate. ALBI, albumin-bilirubin; ATX, autotaxin; HBV, hepatitis B virus; HCV, hepatitis C virus; INR, international normalized ratio; MELD, model for end-stage liver disease. L-ATX, low-ATX group; I-ATX, intermediate-ATX group; H-ATX, high-ATX group.

## Clinical characteristics of the three groups according to baseline serum ATX quartiles

The median baseline ATX level of all patients was 1.377 (1.028–1.763) mg/L (Table 1). The distribution of the low (L)-ATX, intermediate (I)-ATX, and high (H)-ATX groups was 24.8% (52/210), 50.5% (106/210), and 24.8% (52/210), respectively. Significant between-group differences were observed in sex distribution, prevalence of decompensated cirrhosis, hepatic functional reserve-related factors (CP score, MELD score, ALBI score, total bilirubin, albumin, and PT-INR), sodium levels, and platelet count.

## Correlations between serum ATX and baseline characteristics

Serum ATX levels correlated positively with CP score ($r=0.594$), MELD score ($r=0.417$), ALBI score ($r=0.625$), total bilirubin ($r=0.402$), and PT-INR ($r=0.525$), and negatively with albumin ($r=-0.611$), sodium ($r=-0.349$), and platelet count ($r=-0.388$) ($p<0.001$ for all; S2 Table).

## Comparison of cumulative survival rates according to baseline serum ATX quartiles

During the observation period (median, 51.6 [26.6–63.4] months), 43 (20.5%) patients died of liver-related events (liver failure = 32; HCC = 3; rupture of esophageal varices = 5; liver transplantation = 2; and spontaneous bacterial peritonitis = 1).

The 1-, 3-, and 5-year cumulative survival rates were 100%, 100%, and 100% in the L-ATX group; 97.1%, 93.0%, and 88.2% in the I-ATX group; and 80.8%, 55.8%, and 39.5% in the H-ATX group (Fig 1). The cumulative survival rates in the H-ATX group were significantly lower than those in the other two groups ($p<0.001$).

The median baseline ATX levels of male and female patients were 1.189 (0.957–1.607) and 1.560 (1.283–1.890) mg/L, respectively (S1 Fig). Sex-stratified analyses yielded comparable results: the H-ATX $_{male\ (M)}$/H-ATX $_{female\ (F)}$ groups had significantly lower cumulative survival rates than the I-ATX$_M$/I-ATX$_F$ and L-ATX$_M$/L-ATX$_F$ groups ($p<0.001$ for both; S2 Fig).

The median baseline ATX levels of patients with compensated (n = 165) and decompensated cirrhosis (n = 45) were 1.202 (0.971–1.525) and 1.950 (1.594–2.835) mg/L, respectively (S1 Fig). The 1-, 3-, and 5-year cumulative survival rates in the compensated cirrhosis group were 100%, 100%, and 100% in the L-ATX group; 100%, 97.5%, and 95.3% in the I-ATX group; and 100%, 87.0%, and 72.6% in the H-ATX group, indicating that those in the H-ATX group were significantly lower than in the other two groups ($p<0.001$; Fig 2a). Meanwhile, in the decompensated cirrhosis group, no patients were

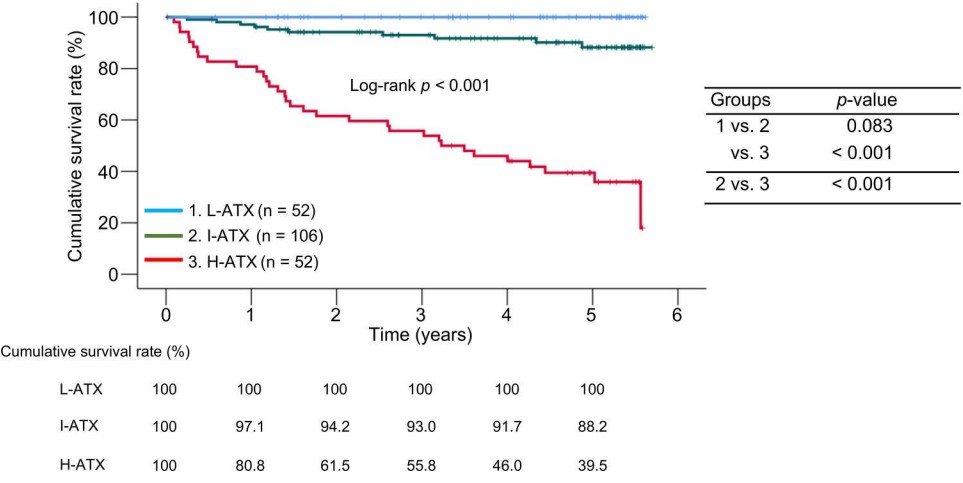

| Cumulative survival rate (%) | | | | | | |
|---|---|---|---|---|---|---|
| L-ATX | 100 | 100 | 100 | 100 | 100 | 100 |
| I-ATX | 100 | 97.1 | 94.2 | 93.0 | 91.7 | 88.2 |
| H-ATX | 100 | 80.8 | 61.5 | 55.8 | 46.0 | 39.5 |

**Fig 1. Comparison of cumulative survival rates among the three groups stratified by quartiles of baseline serum autotaxin (ATX) levels in all patients.** L-ATX, low-ATX group; I-ATX, intermediate-ATX group; H-ATX, high-ATX group.

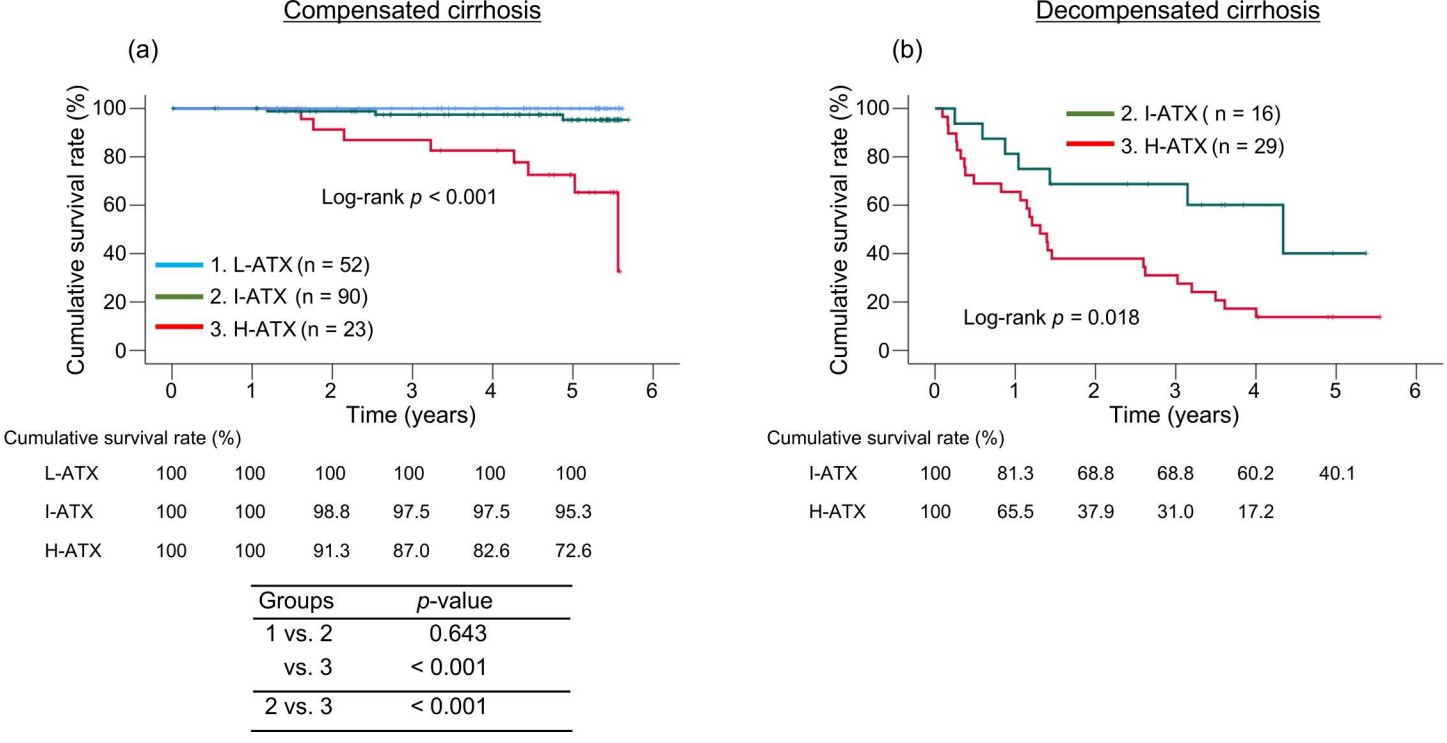

**Fig 2. Comparison of cumulative survival rates among the three groups stratified by quartiles of baseline serum autotaxin (ATX) levels. (a)** Patients with compensated cirrhosis; (b) patients with decompensated cirrhosis. L-ATX, low-ATX group; I-ATX, intermediate-ATX group; H-ATX, high-ATX group.

classified into the L-ATX group; therefore, comparisons were performed between the I-ATX and H-ATX groups. The 1- and 3-year cumulative survival rates were 81.3% and 68.8% in the I-ATX group and 65.5% and 31.0% in the H-ATX group, respectively, with significantly lower survival rates observed in the H-ATX group ($p = 0.018$; Fig 2b).

**Prognostic factors**

On univariate analysis, decompensated cirrhosis, CP score, MELD score, ALBI score, total bilirubin, albumin, PT-INR, creatinine, sodium, platelet count, and ATX were significantly association with mortality in all and male patients; and the same variables except creatinine were significant in female patients (S3 Table). Cox proportional hazards regression analysis identified the following independent factors associated with mortality in all patients (Table 2): decompensated cirrhosis (hazard ratio [HR], 5.486; 95% confidence interval [CI], 2.101–14.324; $p < 0.001$), high ALBI score (HR, 3.040 [1.676–5.514]; $p < 0.001$), low sodium levels (HR, 0.764 [0.672–0.867]; $p < 0.001$), and high ATX levels (HR, 1.661 [1.025–2.690]; $p = 0.039$). In male patients, decompensated cirrhosis, sodium, and ATX remained significant, whereas in female patients, only the CP score remained significant (S4 Table).

When analyzed according to disease stage, univariate analysis identified the following as significant factors associated with mortality: CP score, MELD score, ALBI score, total bilirubin, albumin, PT-INR, sodium, and ATX in patients with compensated cirrhosis; and the same variables were also significant in those with decompensated cirrhosis (S5 Table). Cox proportional hazards regression analysis identified the following independent factors associated with mortality: high ALBI score (HR, 5.682 [1.138–28.362]; $p = 0.034$) and high ATX levels (HR, 7.488 [2.568–21.830]; $p < 0.001$) in patients with

**Table 2. Significant factors associated with mortality in all patients.**

| Variable | Univariate HR (95%CI) | p value | Multivariate HR (95%CI) | p value |
|---|---|---|---|---|
| Gender (Women) | 1.155 (0.630–2.118) | 0.640 | | |
| Age (years) | 1.010 (0.985–1.036) | 0.430 | | |
| Decompensated cirrhosis | 21.427 (10.810–46.527) | < 0.001 | 5.486 (2.101–14.324) | < 0.001 |
| Child-Pugh score | 2.279 (1.953–2.574) | < 0.001 | | |
| MELD score | 1.233 (1.951–2.662) | < 0.001 | | |
| ALBI score | 9.201 (6.029–14.044) | < 0.001 | 3.040 (1.676–5.514) | < 0.001 |
| Sodium (mEq/L) | 0.694 (0.623–0.772) | < 0.001 | 0.764 (0.672–0.867) | < 0.001 |
| Platelet (x10$^4$/µl) | 0.874 (0.816–0.935) | < 0.001 | | |
| Autotaxin (mg/L) | 4.721 (3.361–6.631) | < 0.001 | 1.661 (1.025–2.690) | 0.039 |

ALBI, albumin-bilirubin; CI, confidence interval; HR, hazard ratio; MELD, model for end-stage liver disease.

compensated cirrhosis (Table 3); and high ALBI score (HR, 2.895 [1.583–5.295]; $p$<0.001) and low sodium levels (HR, 0.736 [0.630–0.861]; $p$<0.001) in those with decompensated cirrhosis (S6 Table).

### Comparison of cumulative incidence of decompensation according to baseline serum ATX quartiles

During the observation period, 26 (15.8%) of 165 patients with compensated cirrhosis developed decompensation. The 1-, 3-, and 5-year cumulative incidences of decompensation were 0%, 0%, and 0% in the L-ATX group; 6.8%, 13.0%, and 18.2% in the I-ATX group; and 4.3%, 34.8%, and 49.8% in the H-ATX $_{com}$ group, indicating that the rates in the H-ATX $_{com}$ group were significantly higher than in the other two groups ($p$<0.001; Fig 3).

Sex-specific analyses showed that the H-ATX$_M$ group had the highest cumulative incidence of decompensation in male patients ($p$<0.001). In contrast, among female patients, the cumulative incidence in the H-ATX$_F$ group was comparable to that in the I-ATX$_F$ groups, and both were higher than that in the L-ATX$_F$ group, though not statistically significant ($p$=0.056; S3 Fig).

### Factors associated with decompensation

Univariate analysis identified the following significant factors associated with decompensation development in patients with compensated cirrhosis: CP score, MELD score, ALBI score, total bilirubin, albumin, PT-INR, platelet count, and ATX in all patients; the same variables except MELD score, total bilirubin, and PT-INR in male patients; and the same variables except sodium in female patients (S7 Table). Cox proportional hazards regression analysis identified the following independent factors associated with decompensation development: advanced age (HR, 1.043 [1.005–1.083]; $p$=0.027), high ALBI score (HR, 6.293 [2.084–19.004]; $p$=0.001), low platelet count (HR, 0.883 [0.808–0.966]; $p$=0.006), and high ATX levels (HR, 5.052 [2.354–10.843]; $p$<0.001) in all patients (Table 4). In male patients, ATX was the sole independent factor, whereas in female patients, age, CP score, platelet count, and ATX remained significant (S8 Table).

### Discussion

Cirrhosis is the leading cause of death and disability-adjusted life-years, representing a global public-health burden [1]. Progression from compensated to decompensated cirrhosis drastically worsens prognosis [3]. Therefore, accurately predicting decompensation and intervening early against complications are crucial to improving prognosis. ATX, an established biomarker of liver fibrosis, has been shown to indicate liver disease stage and adverse clinical outcomes in

**Table 3. Significant factors associated with mortality in patients with compensated cirrhosis.**

| Variable | Univariate | | Multivariate | |
|---|---|---|---|---|
| | HR (95%CI) | *p* value | HR (95%CI) | *p* value |
| Gender (Women) | 1.391 (0.424–4.561) | 0.568 | | |
| Age (years) | 1.030 (0.976–1.086) | 0.283 | | |
| Child-Pugh score | 3.391 (1.615–7.119) | 0.001 | | |
| MELD score | 1.243 (0.985–1.568) | 0.067 | | |
| ALBI score | 11.572 (2.920–45.859) | < 0.001 | 5.682 (1.138–28.362) | 0.034 |
| Sodium (mEq/L) | 0.775 (0.620–0.969) | 0.025 | | |
| Autotaxin (mg/L) | 10.862 (4.022–29.334) | < 0.001 | 7.488 (2.568–21.830) | < 0.001 |

ALBI, albumin-bilirubin; CI, confidence interval; HR, hazard ratio; MELD, model for end-stage liver disease.

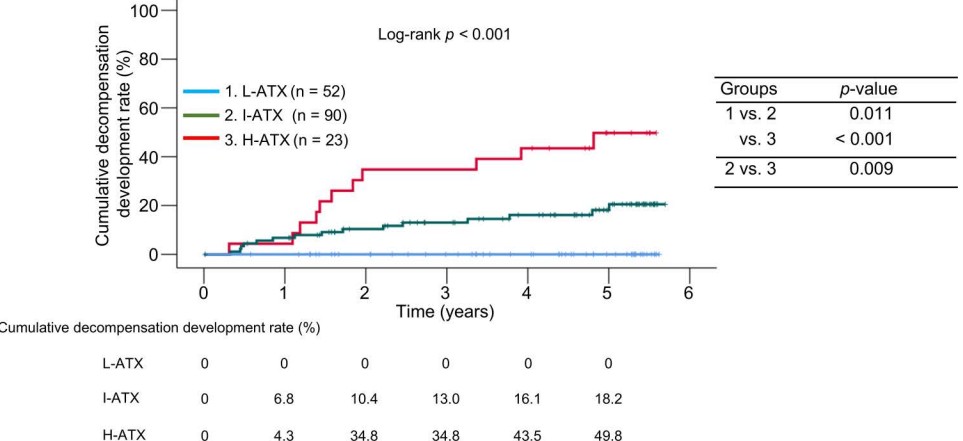

**Fig 3. Comparison of cumulative decompensation incidence rates among the three groups stratified by quartiles of baseline serum autotaxin (ATX) levels in patients with compensated cirrhosis.** L-ATX, low-ATX group; I-ATX, intermediate-ATX group; H-ATX, high-ATX group.

**Table 4. Significant factors associated with decompensation development in all patients.**

| Variable | Univariate | | Multivariate | |
|---|---|---|---|---|
| | HR (95%CI) | *p* value | HR (95%CI) | *p* value |
| Gender (Women) | 1.650 (0.765–3.561) | 0.202 | | |
| Age (years) | 1.027 (0.994–1.062) | 0.114 | 1.043 (1.005–1.083) | 0.027 |
| Child-Pugh score | 3.791 (2.307–6.228) | < 0.001 | | |
| MELD score | 1.165 (0.996–1.363) | 0.056 | | |
| ALBI score | 10.862 (4.386–29.903) | < 0.001 | 6.293 (2.084–19.004) | 0.001 |
| Platelet (x10$^4$/µl) | 0.889 (0.817–0.967) | 0.006 | 0.883 (0.808–0.966) | 0.006 |
| Autotaxin (mg/L) | 6.136 (3.399–11.074) | < 0.001 | 5.052 (2.354–10.843) | < 0.001 |

ALBI, albumin-bilirubin; CI, confidence interval; HR, hazard ratio; MELD, model for end-stage liver disease.

patients with cirrhosis [12–19]. However, the utility of ATX in predicting the progression to decompensation and long-term prognosis in patients with compensated cirrhosis remains unclear. The present study is the first to address these issues. Notably, high serum ATX levels were an independent predictor of mortality in patients with compensated (but not decompensated) cirrhosis, particularly in male patients. Intriguingly, no deaths occurred in any of the L-ATX groups. Furthermore, in patients with compensated cirrhosis, high ATX levels were an independent predictor of decompensation development in the overall cohort and in both male and female populations.

A study of 270 patients with cirrhosis (CP class A, 20.7%; CP class B/C, 79.3%) found that serum ATX levels were significantly correlated with CP class and MELD score; the H-ATX group (> 1.103 mg/L) had poorer survival than the L-ATX group (≤ 1.103 mg/L), and ATX levels were higher in patients with complications such as hepatic encephalopathy or esophageal varices [16]. Another study of 233 patients with cholestatic liver diseases, including PBC (n = 118; cirrhosis, 46.1% [CP class A, 54.7%; CP class B/C, 45.3%]) and PSC (n = 115; cirrhosis, 26.1% [CP class A, 53.3%; CP class B/C, 46.7%]), found a significant correlation between serum ATX activity and disease duration, MELD score, Mayo risk score, and health-related quality of life; increased ATX levels conferred a 4-fold (PBC) and 2.6-fold (PSC) increase in mortality risk [17]. The other study of 309 patients with MASLD (advanced fibrosis [F ≥ 3], 29%) demonstrated that the H-ATX group (≥ 1.227 mg/L) likewise portended inferior survival [18].

The present study recorded the median ATX levels of 1.377 mg/L for the overall cohort, 1.202 mg/L for the compensated cirrhosis group, and 1.950 mg/L for the decompensated cirrhosis group. Multivariable analysis confirmed that high serum ATX levels were an independent predictor of mortality in the overall cohort and compensated cirrhosis group, while high ALBI score and low sodium levels predicted mortality in the decompensated cirrhosis group. These findings suggest that serum ATX levels could serve as a practical and straightforward prognostic biomarker for patients with cirrhosis. However, its utility may diminish once decompensation supervenes, as ATX levels frequently exceed the threshold at which meaningful risk stratification is possible.

Early therapeutic intervention is pivotal to forestall decompensation. A recent meta-analysis identified elevated PT-INR and decreased albumin levels as strong predictors of decompensation, yet these markers lack specificity and are affected by malnutrition (including vitamin K deficiency), malabsorption, and medications [8]. Therefore, biomarkers capturing the various pathophysiological mechanisms that drive decompensation are desirable. Notably, this study demonstrated that ATX outperformed the CP score, which includes PT-INR and albumin, for predicting decompensation in patients with compensated cirrhosis, particularly in male patients. The cumulative incidence of decompensation was significantly higher in the H-ATX group than in the I-ATX and L-ATX groups, with nopatients in the L-ATX group progressing to decompensation. A study of 190 patients with PBC found that H-ATX group (≥ 1.086 mg/L) had a significantly higher incidence of liver-related events (LREs), including ascites, hepatic encephalopathy, esophagogastric varices, and HCC, than the L-ATX group (< 1.086 mg/L) [19]. Additionally, serum ATX levels showed an independent association with LRE incidence. The aforementioned study on patients with MASLD also identified high serum ATX levels as an independent predictor of LREs [18]. Hepatic venous pressure gradient (HVPG), a well-established indicator of portal hypertension (PH), is the most reliable predictor of decompensation, with a greater discriminative ability than CP and MELD scores [25]. Patients with an HVPG < 10 mm Hg showed a 90% probability of not developing decompensation over a median follow-up of 4 years. Given that HVPG measurements are invasive and require expertise in interventional radiology, repeated evaluation is impractical. In contrast, serum ATX offers a non-invasive, readily repeatable surrogate that could be integrated into longitudinal monitoring, potentially in combination with other parameters to enhance predictive accuracy, which presents a challenge for future research.

Mechanistically, circulating ATX is cleared by hepatic sinusoidal endothelial cells; therefore, endothelial dysfunction in cirrhosis leads to elevated serum ATX levels [11,12]. LPA, which is derived from ATX-mediated hydrolysis of lysophosphatidylcholine, promotes the proliferation and contractility of hepatic stellate cells [10,11]. Progression to PH is attributed to increased intrahepatic vascular resistance to portal blood flow, hepatic stellate cell activation, and endothelial cell

dysfunction, culminating in complications such as collateral formation, portosystemic shunt, and ascites [16,26]. These hemodynamic changes eventually lead to variceal bleeding, hyperammonemia, and hepatic encephalopathy. Therefore, serum ATX levels may reflect the degree of PH. An *in vivo* study using a thioacetamide-induced acute hepatic encephalopathy model demonstrated that inhibiting ATX reduced plasma LPA and ammonia levels, restored impaired blood–brain barrier permeability, and improved both recognition memory and hepatic encephalopathy stage [27]. These findings indicate the involvement of the ATX–LPA axis in the pathogenesis of hepatic encephalopathy and support a pathophysiological link between the ATX–LPA axis and decompensation.

In this study, serum ATX levels were significantly higher in female patients than in male patients, consistent with previous reports [18,24]. Therefore, sex-specific cutoff values were applied in the stratified analyses. Although the mechanisms underlying this sex difference remain unclear, previous studies suggest that ATX expression may be influenced by sex steroids, including estrogen [18,28]. Accordingly, this sex-related difference should be considered when using ATX as a biomarker for decompensation and prognosis in patients with cirrhosis.

This study has some limitations. First, this was a retrospective, two-center study; therefore, prospective, multi-center studies are needed to obtain more definitive evidence. Second, serial ATX measurements were unavailable, precluding assessment of the relationship between longitudinal changes in serum ATX levels and decompensation development or prognosis. Finally, HVPG, the gold standard for portal pressure assessment, was not measured, leaving its correlation with serum ATX levels unresolved.

## Conclusions

High serum ATX levels are strongly associated with the development of decompensation and poor long-term prognosis in patients with compensated cirrhosis. Thus, ATX represents a promising, non-invasive surrogate biomarker for risk stratification and may aid in guiding personalized management strategies in this population.

## Supporting information

**S1 Fig. Comparison of serum autotaxin levels according to sex and cirrhosis stage.** (a) Male vs. female patients; (b) Compensated vs. decompensated cirrhosis. The numbers indicate the 3rd quartile, median, and 1st quartile, respectively, from top to bottom.
(TIF)

**S2 Fig. Comparison of cumulative survival rates among the three groups stratified by quartiles of baseline serum autotaxin (ATX) levels.** (a) Male patients; (b) female patients. L-ATX$_{men (M)/women (W)}$, low-ATX group; I-ATX$_{M/W}$, intermediate-ATX group; H-ATX$_{M/W}$, high-ATX group.
(TIF)

**S3 Fig. Comparison of cumulative incidence of decompensation among the three groups stratified by quartiles of baseline serum autotaxin (ATX) levels in patients with compensated cirrhosis.** (a) Male patients; (b) female patients. L-ATX$_{men (M)/women (W)}$, low-ATX group; I-ATX$_{M/W}$, intermediate-ATX group; H-ATX$_{M/W}$, high-ATX group.
(TIF)

**S1 Table. Clinical characteristics by gender.**
(DOCX)

**S2 Table. Correlation between serum autotaxin levels and baseline characteristics.**
(DOCX)

**S3 Table. Univariate analysis of factors associated with mortality in all patients and according to sex.**
(DOCX)

**S4 Table. Significant factors associated with mortality according to sex.**
(DOCX)

**S5 Table. Univariate analysis of factors associated with mortality in patients with compensated and decompensated cirrhosis.**
(DOCX)

**S6 Table. Significant factors associated with mortality in patients with decompensated cirrhosis.**
(DOCX)

**S7 Table. Univariate analysis of factors associated with decompensation development in all patients and according to sex.**
(DOCX)

**S8 Table. Significant factors associated with decompensation development according to sex.**
(DOCX)

## Author contributions

**Conceptualization:** Chisato Saeki.

**Data curation:** Chisato Saeki.

**Formal analysis:** Chisato Saeki.

**Investigation:** Chisato Saeki, Tsunekazu Oikawa, Tomoya Kanai, Sachie Kiryu, Hiroshi Kamioka, Kaoru Ueda, Masanori Nakano, Yuichi Torisu.

**Methodology:** Chisato Saeki, Akihito Tsubota.

**Project administration:** Chisato Saeki.

**Supervision:** Masayuki Saruta, Akihito Tsubota.

**Validation:** Chisato Saeki, Tsunekazu Oikawa, Yuichi Torisu, Akihito Tsubota.

**Visualization:** Chisato Saeki.

**Writing – original draft:** Chisato Saeki, Akihito Tsubota.

**Writing – review & editing:** Masayuki Saruta, Akihito Tsubota.

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
