## [Decision Letter · Decision Letter 0]

5 Mar 2026

Dear Dr. Saeki,

Thank you for submitting your manuscript to PLOS ONE. After careful consideration, we feel that it has merit but does not fully meet PLOS ONE’s publication criteria as it currently stands. Therefore, we invite you to submit a revised version of the manuscript that addresses the points raised during the review process.

We look forward to receiving your revised manuscript.

Kind regards,

Pei-Yuan Su

Academic Editor

PLOS One

Journal Requirements:

https://journals.plos.org/plosone/s/file?id=ba62/PLOSOne_formatting_sample_title_authors_affiliations.pdf....

Please update your Data Availability statement in the submission form accordingly

Reviewers' comments:

Reviewer's Responses to Questions

**Comments to the Author**

1. Is the manuscript technically sound, and do the data support the conclusions?

Reviewer #1: No

Reviewer #2: Yes

2. Has the statistical analysis been performed appropriately and rigorously?

Reviewer #1: No

Reviewer #2: Yes

3. Have the authors made all data underlying the findings in their manuscript fully available?

Reviewer #1: Yes

Reviewer #2: Yes

4. Is the manuscript presented in an intelligible fashion and written in standard English?

Reviewer #1: Yes

Reviewer #2: Yes

Reviewer #1: In this manuscript, Saeki et al. investigated the clinical utility of serum autotaxin (ATX) levels as a predictive biomarker for decompensation and mortality, particularly in patients with compensated cirrhosis. They concluded that ATX represents a promising non-invasive biomarker for predicting decompensation and poor prognosis in patients with compensated cirrhosis. Multivariate analysis identified high serum ATX levels as an independent predictor of mortality and an increased risk of decompensation. However, there were some critical issues that should be addressed.

Major

1. This study concluded that high ATX levels can predict prognosis and decompensation of compensated cirrhosis. However, this was an analysis of a very heterogeneous population, including patients with different underlying cirrhosis diseases and those with hepatocellular carcinoma. Due to the different treatments for the underlying liver disease, there are many confounding factors related to prognosis and liver function, making the present conclusions difficult to generalize.

2. In the prognostic analysis, age and sex were not included in the Cox proportional hazards analysis. Regardless of the univariate results, they should be included as factors, and if not, the reason should be clearly stated.

3. In Table 1, all cases are divided into three groups based on the quartiles of ATX. However, other analyses have yielded different ranges than those in Table 1, which can be confusing. Furthermore, having multiple classification criteria makes it difficult to use in general practice. Furthermore, ATX levels are higher in women than in men, and this should be considered in the main analysis.

Minor

1. Some of the supplementary tables contain nearly identical data and should be combined.

2. Follow-up procedures (e.g., intervals and techniques of examinations) should be described.

Reviewer #2: Clinical usefulness of serum autotaxin levels for predicting decompensation development and prognosis in patients with compensated cirrhosis.” by Saeki et al.

This study demonstrates that elevated serum autotaxin (ATX) levels are strongly associated with decompensation and overall prognosis in patients with cirrhosis, particularly compensated cirrhosis. The relatively large cohort (n = 258) and the detailed statistical analyses and discussion are commendable, and the overall findings are generally acceptable and clinically meaningful.

However, I would like to raise the following concerns.

Major Comment

In this study, 48 patients (18.6%) with early-stage hepatocellular carcinoma (HCC; BCLC stage 0 or A) were included in the cohort. In the univariate and multivariate analyses presented in Table 2 and Supplementary Tables S3–10, the presence of HCC was not identified as a significant factor affecting overall survival. However, in the subsequent multivariable analysis (Table 4), the presence of HCC was extracted as an independent predictor of decompensation.

In general, concomitant HCC is one of the most influential determinants of both prognosis and hepatic decompensation in patients with cirrhosis. Even in patients with early-stage HCC (BCLC 0 or A), curative interventions such as surgical resection or radiofrequency ablation may adversely affect hepatic functional reserve and thereby increase the risk of decompensation. Therefore, the presence of HCC—even at an early stage—may act as a significant confounding factor when evaluating the association between serum ATX levels and clinical outcomes, including both overall survival and decompensation.

For this reason, it would be more appropriate either (1) to exclude patients with concomitant HCC from the primary analysis, or (2) to perform stratified analyses according to HCC status (HCC vs. non-HCC) to clarify whether the prognostic value of serum ATX remains consistent in patients without HCC. Without such analyses, the independent predictive value of ATX may be difficult to interpret.

Minor Comment

In the univariate and multivariate analyses, ALBI score was included in the models together with total bilirubin and albumin as separate covariates. However, because the ALBI score is calculated directly from serum albumin and total bilirubin levels, including these variables simultaneously in the same regression model may introduce multicollinearity and compromise the interpretability and stability of the model estimates.

It would be more appropriate to construct multivariable models using either the ALBI score or its individual components (albumin and total bilirubin), but not both simultaneously. A similar methodological concern applies, strictly speaking, to composite scores such as Child–Pugh or MELD when their components are included in the same model.

.

Reviewer #1: No

Reviewer #2: No

---

## [Author Response · Author response to Decision Letter 1]

9 Mar 2026

RESPONSES TO THE REVIEWER

We appreciate the opportunity to submit a revised version of our manuscript entitled “Clinical usefulness of serum autotaxin levels for predicting decompensation development and prognosis in patients with compensated cirrhosis” to PLOS One. We also appreciate the reviewers' constructive comments and suggestions. Our point-by-point responses to the reviewers’ comments are listed below.

Reviewer Comments:

Reviewer #1

In this manuscript, Saeki et al. investigated the clinical utility of serum autotaxin (ATX) levels as a predictive biomarker for decompensation and mortality, particularly in patients with compensated

cirrhosis. They concluded that ATX represents a promising non-invasive biomarker for predicting decompensation and poor prognosis in patients with compensated cirrhosis. Multivariate analysis identified high serum ATX levels as an independent predictor of mortality and an increased risk of decompensation. However, there were some critical issues that should be addressed.

Major

1.This study concluded that high ATX levels can predict prognosis and decompensation of compensated cirrhosis. However, this was an analysis of a very heterogeneous population, including patients with different underlying cirrhosis diseases and those with hepatocellular carcinoma. Due to the different treatments for the underlying liver disease, there are many confounding factors related to prognosis and liver function, making the present conclusions difficult to generalize.

Responses: We are thankful for the reviewer’s critical comments. As pointed out by the reviewer, the study population was heterogeneous, particularly with the inclusion of patients with HCC (BCLC stage 0 or A). These patients underwent curative treatments such as surgical resection or radiofrequency ablation, as well as transarterial chemoembolization (TACE), which may substantially affect hepatic functional reserve, HCC recurrence, the risk of decompensation, and prognosis. Indeed, the presence of HCC was identified as a significant independent predictor of decompensation. Therefore, we reanalyzed all data after excluding patients with HCC (n = 210) and regenerated all figures and tables. After reanalysis, the key findings remained unchanged; specifically, ATX was a significant independent predictor of mortality and decompensation in patients with compensated cirrhosis (new Tables 3 and 4).

2. In the prognostic analysis, age and sex were not included in the Cox proportional hazards analysis. Regardless of the univariate results, they should be included as factors, and if not, the reason should be

clearly stated.

Responses: We are thankful for the reviewer’s constructive suggestions. As pointed out by the reviewer, age and sex are clinically relevant covariates, and we therefore included them in all multivariate Cox proportional hazards analyses regardless of their univariate significance. We have added these descriptions to the Methods section (lines 159–160 in the revised manuscript).

3. In Table 1, all cases are divided into three groups based on the quartiles of ATX. However, other analyses have yielded different ranges than those in Table 1, which can be confusing. Furthermore, having

multiple classification criteria make it difficult to use in general practice. Furthermore, ATX levels are higher in women than in men, and this should be considered in the main analysis.

Responses: We are thankful for the reviewer’s constructive suggestions. To classify patients into three groups, we set a total of seven cutoff values, including sex- or stage-specific thresholds; however, as pointed out by the reviewer, this approach is quite complex and may limit its clinical applicability. Given that ATX levels are higher in females than in males, we performed all analyses using three cutoff values: a common cutoff value and sex-specific cutoff values (Fig. S1).

Minor

1. Some of the supplementary tables contain nearly identical data and

should be combined.

Responses: We are thankful for the reviewer’s comments. Because the large number of supplementary tables made the results difficult to follow, we combined several tables and reorganized them to improve clarity (e.g., Tables S3, S4, and S5 were combined into a new Table S3).

2. Follow-up procedures (e.g., intervals and techniques of examinations)

should be described.

Responses: We are thankful for the reviewer’s comments. We followed patients regularly in the outpatient clinics every 1–2 months and performed laboratory tests at each visit. We conducted imaging examinations, including computed tomography, magnetic resonance imaging, and/or ultrasonography, approximately every 4 months to assess ascites and HCC, and performed endoscopic examinations for variceal surveillance every 6–12 months. We have added these descriptions to the Methods section (lines 119–123 in the revised manuscript).

Reviewer #2

Clinical usefulness of serum autotaxin levels for predicting decompensation development and prognosis in patients with compensated cirrhosis.” by Saeki et al. This study demonstrates that elevated serum autotaxin (ATX) levels are strongly associated with decompensation and overall prognosis in patients with cirrhosis, particularly compensated cirrhosis. The relatively large cohort (n = 258) and the detailed statistical analyses

and discussion are commendable, and the overall findings are generally acceptable and clinically meaningful. However, I would like to raise the following concerns.

Major Comment

1. In this study, 48 patients (18.6%) with early-stage hepatocellular carcinoma (HCC; BCLC stage 0 or A) were included in the cohort. In the univariate and multivariate analyses presented in Table 2 and

Supplementary Tables S3–10, the presence of HCC was not identified as a significant factor affecting overall survival. However, in the subsequent multivariable analysis (Table 4), the presence of HCC was

extracted as an independent predictor of decompensation.

In general, concomitant HCC is one of the most influential determinants of both prognosis and hepatic decompensation in patients with cirrhosis. Even in patients with early-stage HCC (BCLC 0 or A), curative

interventions such as surgical resection or radiofrequency ablation may adversely affect hepatic functional reserve and thereby increase the risk of decompensation. Therefore, the presence of HCC—even at an early

stage—may act as a significant confounding factor when evaluating the association between serum ATX levels and clinical outcomes, including both overall survival and decompensation.

For this reason, it would be more appropriate either (1) to exclude patients with concomitant HCC from the primary analysis, or (2) to perform stratified analyses according to HCC status (HCC vs. non-HCC) to

clarify whether the prognostic value of serum ATX remains consistent in patients without HCC. Without such analyses, the independent predictive value of ATX may be difficult to interpret.

Responses: We are thankful for the reviewer’s critical comments. The study population included patients with HCC (BCLC stage 0 or A). In fact, these patients underwent treatments such as surgical resection, radiofrequency ablation, or transarterial chemoembolization (TACE), which can affect hepatic functional reserve, HCC recurrence, the development of decompensation, and prognosis. As pointed out by the reviewer, the presence of HCC was identified as a significant independent predictor of decompensation, suggesting a potential impact of HCC on the development of decompensation. To address this issue, we excluded patients with HCC (n = 210), performed all analyses again, and reconstructed all figures and tables. As a result, the main findings were unchanged, and ATX remained a significant independent predictor of mortality and decompensation in patients with compensated cirrhosis (new Tables 3 and 4).

Minor Comment

1. In the univariate and multivariate analyses, ALBI score was included in the models together with total bilirubin and albumin as separate covariates. However, because the ALBI score is calculated directly from

serum albumin and total bilirubin levels, including these variables simultaneously in the same regression model may introduce multicollinearity and compromise the interpretability and stability of the model estimates.

It would be more appropriate to construct multivariable models using either the ALBI score or its individual components (albumin and total bilirubin), but not both simultaneously. A similar methodological

concern applies, strictly speaking, to composite scores such as Child–Pugh or MELD when their components are included in the same model.

Responses: We are thankful for the reviewer’s critical comments. As suggested by the reviewer, when performing multivariate analyses, albumin, total bilirubin, PT-INR, and creatinine were excluded if they were simultaneously significant in the univariate analysis together with the Child–Pugh, MELD, or ALBI scores to avoid multicollinearity. We have added these descriptions to the Methods section (lines 160–163 in the revised manuscript).

---

## [Decision Letter · Decision Letter 1]

23 Mar 2026

Dear Dr. Saeki,

Thank you for submitting your manuscript to PLOS ONE. After careful consideration, we feel that it has merit but does not fully meet PLOS ONE’s publication criteria as it currently stands. Therefore, we invite you to submit a revised version of the manuscript that addresses the points raised during the review process.

We look forward to receiving your revised manuscript.

Kind regards,

Pei-Yuan Su

Academic Editor

PLOS One

Journal Requirements:

Reviewers' comments:

Reviewer's Responses to Questions

**Comments to the Author**

Reviewer #1: (No Response)

Reviewer #2: All comments have been addressed

2. Is the manuscript technically sound, and do the data support the conclusions?

Reviewer #1: Yes

Reviewer #2: Yes

3. Has the statistical analysis been performed appropriately and rigorously?

Reviewer #1: Yes

Reviewer #2: Yes

4. Have the authors made all data underlying the findings in their manuscript fully available?

The PLOS Data policy requires authors to make all data underlying the findings described in their manuscript fully available without restriction, with rare exception (please refer to the Data Availability Statement in the manuscript PDF file). The data should be provided as part of the manuscript or its supporting information, or deposited to a public repository. For example, in addition to summary statistics, the data points behind means, medians and variance measures should be available. If there are restrictions on publicly sharing data—e.g. participant privacy or use of data from a third party—those must be specified. requires authors to make all data underlying the findings described in their manuscript fully available without restriction, with rare exception (please refer to the Data Availability Statement in the manuscript PDF file). The data should be provided as part of the manuscript or its supporting information, or deposited to a public repository. For example, in addition to summary statistics, the data points behind means, medians and variance measures should be available. If there are restrictions on publicly sharing data—e.g. participant privacy or use of data from a third party—those must be specified.

Reviewer #1: Yes

Reviewer #2: Yes

5. Is the manuscript presented in an intelligible fashion and written in standard English?

Reviewer #1: Yes

Reviewer #2: Yes

Reviewer #1: The authors addressed most of the reviewers' concerns. However, there is disagreement regarding the ATX cutoff values. While the text states that "all analyses were stratified by sex using sex-specific cutoff values. Patients were categorized into three groups according to serum ATX levels," Table 1 categorizes all case data into three groups instead of separating by sex. Therefore, for example, it is unclear whether a female case with an ATX of 1.8 is classified as Intermediate or High. The data should be standardized to sex-specific cutoff values as stated in the text. Furthermore, the authors should discuss why ATX levels are higher in female than in male.

Reviewer #2: The authors have adequately addressed the issues I raised, and the manuscript has been substantially improved. In its current form, I consider it acceptable for publication.

**Do you want your identity to be public for this peer review?** For information about this choice, including consent withdrawal, please see our  For information about this choice, including consent withdrawal, please see our  For information about this choice, including consent withdrawal, please see our  For information about this choice, including consent withdrawal, please see our Privacy Policy..

Reviewer #1: No

Reviewer #2: No

---

## [Author Response · Author response to Decision Letter 2]

26 Mar 2026

RESPONSES TO THE REVIEWER

We appreciate the opportunity to submit a revised version of our manuscript entitled “Clinical usefulness of serum autotaxin levels for predicting decompensation development and prognosis in patients with compensated cirrhosis” to PLOS One. We also appreciate the reviewers' constructive comments and suggestions. Our point-by-point responses to the reviewers’ comments are listed below.

Reviewer Comments:

Reviewer #1

The authors addressed most of the reviewers' concerns. However, there is disagreement regarding the ATX cutoff values. While the text states that "all analyses were stratified by sex using sex-

specific cutoff values. Patients were categorized into three groups according to serum ATX levels," Table 1 categorizes all case data into three groups instead of separating by sex. Therefore, for example, it is unclear whether a female case with an ATX of 1.8 is classified as Intermediate or High. The data should be standardized to sex-specific cutoff values as stated in the text. Furthermore, the authors should discuss why ATX levels are higher in female than in male.

Responses: We apologize that the description of the patient classification in the Methods section may have been misleading. First, we established cutoff values based on the overall cohort and performed stratified analyses. Subsequently, given that sex differences in ATX levels have been reported, we also performed sex-stratified analyses using sex-specific cutoff values. We believe that this stepwise approach, from analyses of the overall cohort to sex-stratified analyses, is reasonable and methodologically appropriate. We have revised the description of the patient classification in the Methods section (lines 134–137 in the revised manuscript).

In this study, serum ATX levels were significantly higher in female patients than in male patients, in line with previous reports [Refs. 18, 25]. Although the mechanisms underlying this sex difference remain unclear, previous studies have suggested that ATX expression may be influenced by sex steroids, including estrogen (Refs. 18, 29). Thus, this difference may need to be considered when using ATX as a biomarker for decompensation and prognosis. We have added these descriptions to the Discussion section (lines 398–394 in the revised manuscript).

Reviewer #2

The authors have adequately addressed the issues I raised, and the manuscript has been substantially improved. In its current form, I consider it acceptable for publication.

Responses: We are grateful for the reviewer’s encouraging comments.

---

## [Editor Report · Decision Letter 2]

1 Apr 2026

Clinical usefulness of serum autotaxin levels for predicting decompensation development and prognosis in patients with compensated cirrhosis

PONE-D-25-67582R2

Dear Dr. Saeki,

We’re pleased to inform you that your manuscript has been judged scientifically suitable for publication and will be formally accepted for publication once it meets all outstanding technical requirements.

Kind regards,

Pei-Yuan Su

Academic Editor

PLOS One
---

## [Editor Report · Acceptance letter]

PONE-D-25-67582R2

PLOS One

Dear Dr. Saeki,

I'm pleased to inform you that your manuscript has been deemed suitable for publication in PLOS One. Congratulations! Your manuscript is now being handed over to our production team.

Kind regards,

on behalf of

Dr. Pei-Yuan Su

Academic Editor

PLOS One